# Rational enzyme design for enabling biocatalytic Baldwin cyclization and asymmetric synthesis of chiral heterocycles

Jun-Kuan Li[1,2,5], Ge Qu[2,3,5], Xu Li[2,3,5], Yuchen Tian[1], Chengsen Cui[2,3], Fa-Guang Zhang[1], Wuyuan Zhang[2,3], Jun-An Ma[1]✉, Manfred T. Reetz[2,4]✉ & Zhoutong Sun[2,3]✉

Chiral heterocyclic compounds are needed for important medicinal applications. We report an in silico strategy for the biocatalytic synthesis of chiral *N*- and *O*-heterocycles via Baldwin cyclization modes of hydroxy- and amino-substituted epoxides and oxetanes using the limonene epoxide hydrolase from *Rhodococcus erythropolis*. This enzyme normally catalyzes hydrolysis with formation of vicinal diols. Firstly, the required shutdown of the undesired natural water-mediated ring-opening is achieved by rational mutagenesis of the active site. In silico enzyme design is then continued with generation of the improved mutants. These variants prove to be versatile catalysts for preparing chiral *N*- and *O*-heterocycles with up to 99% conversion, and enantiomeric ratios up to 99:1. Crystal structural data and computational modeling reveal that Baldwin-type cyclizations, catalyzed by the reprogrammed enzyme, are enabled by reshaping the active-site environment that directs the distal RHN and HO-substituents to be intramolecular nucleophiles.

The Baldwin cyclization rules were originally proposed with the aim of systematizing a huge set of previous empirical observations[1,2]. Baldwin proposed that maximum orbital overlap as the crucial element of stereoelectronic effects is the reason for the often observed regioselectivity leading to differently sized cyclic products. In seminal prior work, Eschenmoser et al. studied ring closures involving $S_N2$ reactions at $sp^3$ hybridized C-centers, demonstrating that they occur most rapidly when an angle of 180° between the three interacting atoms is maintained[3]. They also surmised that when the intramolecular substitution reaction occurs at the C-atoms of epoxides, the extent of the stereoelectronic effect may be somewhat relaxed. These developments inspired Janda, Lerner and coworkers to generate catalytic antibodies which catalyze a disfavored ring closure reaction (anti-Baldwin) of a specifically designed epoxy alcohol[4,5]. A chiral *N*-oxide antigen was coupled to the keyhole limpet hemocyanin, the conjugate then being used to immunize 129 mice for monoclonal antibody

production. It was found that the 6-endo-tet cyclization mode with formation of the anti-Baldwin tetrahydropyran product dominates. In the absence of the antibody, cyclization to the normal product occurred, a background reaction in the designed system. Nicolaou et al. described an unusual anti-Baldwin cyclization in the synthesis of brevetoxin B[6]. More recently, Alabugin et al. have analyzed the Baldwin rules more closely and extended them accordingly[7,8]. They pointed out, inter alia, that in both cyclization modes of the Janda/Lerner-study, the "breaking bond is outside of the formed ring", calling for an extension of the original Baldwin rules in which spiro and fused transition states, respectively, are compared[8]. Relevant is the literature describing polyether biosynthesis in which both favored and disfavored epoxy-alcohol ring closures occur. Hotta, Chen, Houk and coworkers have studied these enzymatic processes by X-ray structures and quantum mechanical (QM) computations[9]. In this context, other Baldwin cyclization studies deserve mention[10,11].

[1]Department of Chemistry, Frontiers Science Center for Synthetic Biology (Ministry of Education), Tianjin University, Tianjin 300072, China. [2]Tianjin Institute of Industrial Biotechnology, Chinese Academy of Sciences, Tianjin 300308, China. [3]National Technology Innovation Center of Synthetic Biology, Tianjin 300308, China. [4]Biocatalysis Section, Max-Planck-Institut für Kohlenforschung, Kaiser-Wilhelm-Platz 1, 45470 Mülheim an der Ruhr, Germany. [5]These authors contributed equally: Jun-Kuan Li, Ge Qu, Xu Li. ✉e-mail: majun_an68@tju.edu.cn; reetz@mpi-muelheim.mpg.de; sunzht@tib.cas.cn

In the present study, we devised a rational biocatalytic mutagenesis strategy using an epoxide hydrolase as the enzyme and hydroxyl/amino-substituted epoxides and oxetanes as substrates, the aim being to induce Baldwin cyclization rather than hydrolysis leading to the respective natural hydrolytic 1,2-diol products (Fig. 1a). Control of Baldwin and anti-Baldwin cyclization modes was part of the challenge. Originally, we chose 1-epoxy-1-pentanol as the model substrate, structurally similar to the substrate in the Janda/Lerner study, but soon discovered that even in the absence of an enzyme this compound undergoes a rapid background reaction leading to the respective tetrahydrofuran derivative. Rather than subtracting the effect of the background reaction from the experimentally observed result[4], we opted for the next higher homologous substrate 4-(oxiran-2-yl)butan-1-ol with formation of tetrahydropyran-2-methanol as Baldwin product and oxepan-3-ol as anti-Baldwin product (Fig. 1b). Limonene epoxide hydrolase (LEH) was chosen as the enzyme, its crystal structure and mechanism being essential for our endeavor. We were especially motivated by the possibility to access a broad range of N- and O-heterocyclic building blocks needed for the synthesis of chiral therapeutic drugs, including anti-cancer compounds, antibiotics and immunosupressants (Supplementary Fig. 1)[12,13].

Epoxides and oxetanes have been used as starting materials for chemical syntheses of chiral heterocycles by employing chiral transition metal catalysts or reagents[14–17]. In this realm, the Jacobsen group[14] reported the intramolecular enantioselective ring opening of epoxides for the preparation of chiral O-heterocycles synthesis. Intramolecular desymmetrization of oxetanes has also been established as a synthetically useful strategy[16]. Nevertheless, most of these approaches require either transition metals, which in the workup need to be eliminated, or stoichiometric amounts of organic reagents. It inevitably leads to extra operational labor or stoichiometric waste generation, so the development of economically and ecologically viable catalytic approaches remains in high demand. Enzymes are alternatives, which can catalyze reactions with unrivaled rate and exceptional selectivity[18–21].

Nature has created a vast repertoire of enzymes that synthesize chiral N- and O-heterocycles through ring-opening and intramolecular transformations[9,22], including epoxidation-cyclization, chlorination-cyclization and others, but the respective substrate scopes are limited, as in polyether epoxide hydrolases/cyclases and isomerases/cyclases[23,24]. Consequently, we wished to develop an alternative biocatalytic approach, in which a single catalyst platform would provide direct access to diverse chiral heterocycles from appropriate epoxide/oxetane precursors as starting materials. It should be noted that oxetanes have never been subjected to any epoxide hydrolase-mediated transformations.

We turned to cofactor-independent limonene epoxide hydrolases (LEHs) that normally catalyze direct hydrolysis of epoxides by positioning and activating water at the active site. Taking the LEH from *Rhodococcus erythropolis* (*Re*LEH) as an example, it is highly selective to the hydrolysis of the epoxide substrates with the formation of the respective chiral vicinal diols, which in our endeavor had to be shut down, as will be shown. On the basis of the well-documented mechanism of *Re*LEH, a water molecular is positioned by H-bonds to N55 and Y53 while D132 abstracts a proton from the water, enabling a smooth nucleophilic attack on one of the substrate's epoxide carbon center (Fig. 1a)[25]. In addition, previous studies have revealed that some EHs convert bis-epoxy polyketide

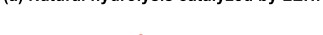

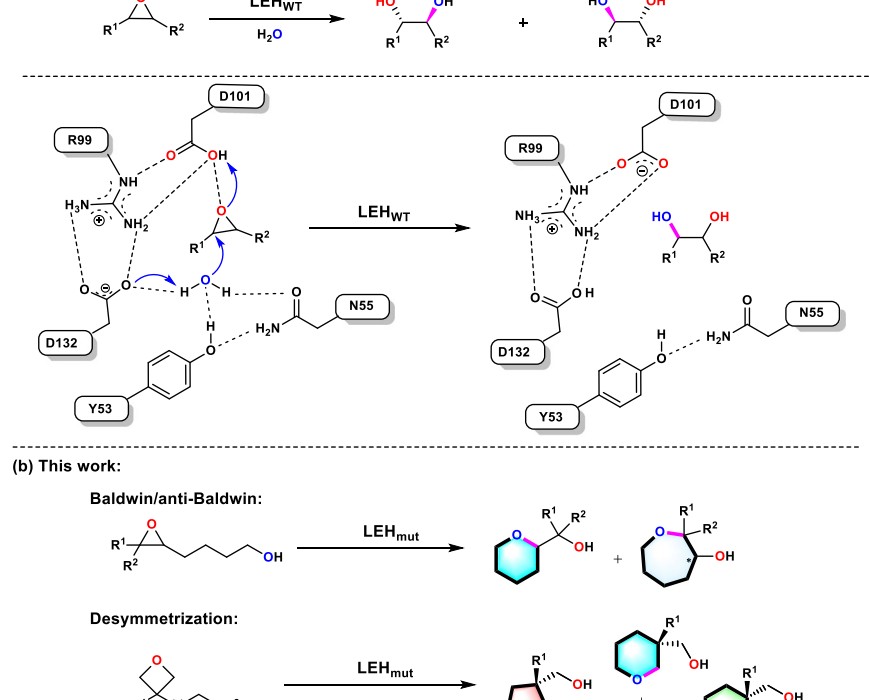

**Fig. 1 | Schematic representation of LEH-catalyzed reactions. a** Natural hydrolytic reaction catalyzed by wild-type (WT) *Re*LEH (top) and its known catalytic mechanism (bottom)[23]; D, Aspartic acid; N, Asparagine; Y, Tyrosine. **b** The proposed approach to biocatalytic routes using rationally engineered *Re*LEH mutants in this work.

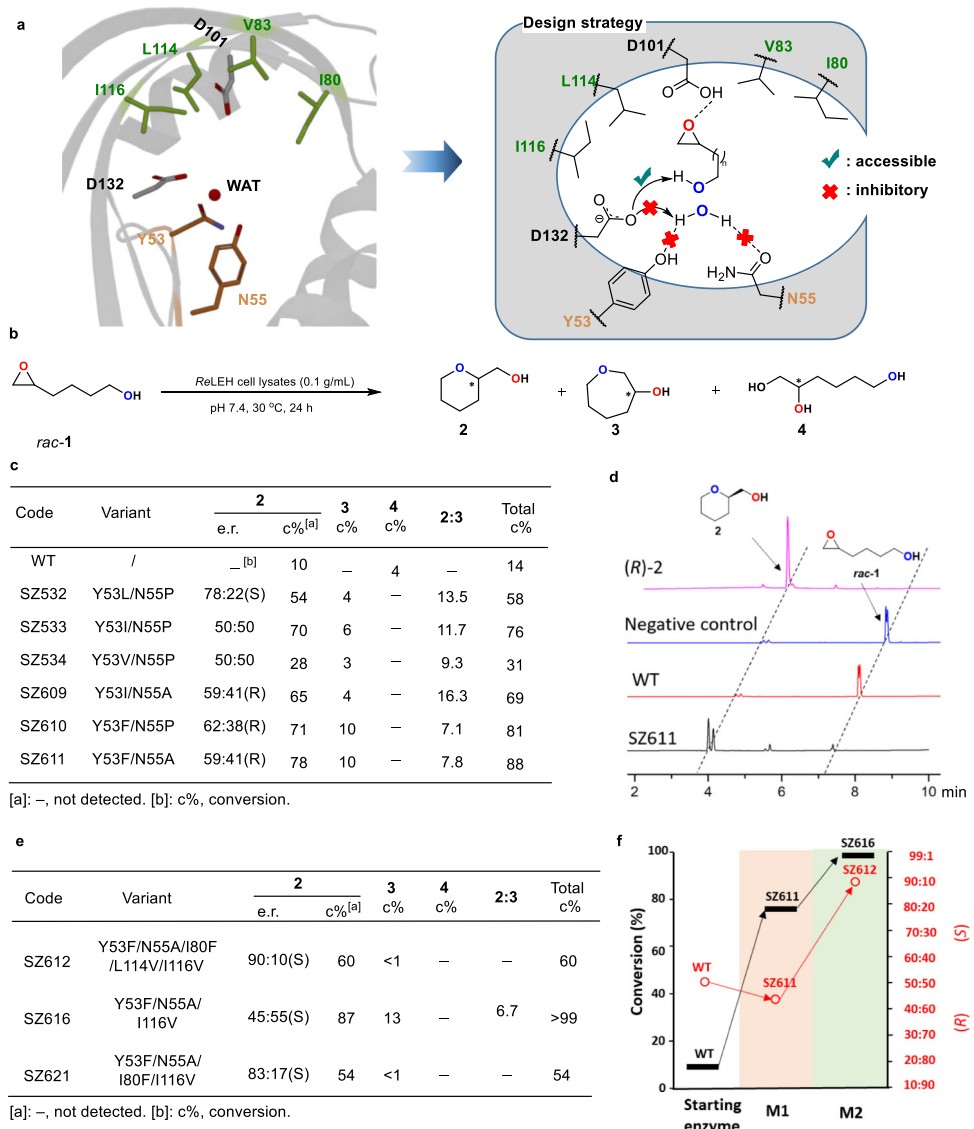

**Fig. 2 | Reprogramming the catalytic repertoire of *Re*LEH towards the Baldwin-type reactions. a** Design strategy of breaking the water binding network in *Re*LEH. **b** *Re*LEH-catalyzed model reaction leading to three distinct products. **c** Screening results of the first round of mutagenesis performed on residues 53 and 55. **d** GC profiles of the mutant SZ611. **e** Screening results of the second round of mutagenesis targeting residues 80, 83, 114 and 116. **f** Conversion and e.r. values of the mutants. M1 and M2 indicate the first and the second rounds of mutagenesis. Reaction conditions: LEH (0.1 g/mL wet cell), *rac*-1 (5 mM), PBK buffer (50 mM, pH 7.4), Lysozyme (1 mg/mL), DNase I (6 U/mL), 0.4 mL total volume, 30 °C, 1000 rpm, 24 h.

intermediates into the corresponding furan-/pyran-cyclized natural products, and the stereochemical courses of the cyclization were solely governed by Baldwin's rules[10,26–30]. In contrast, the enzymatic processes in Fig. 1b catalyzed by LEH have not been reported to date. As a first step in achieving this reprogrammed reactivity, the natural hydrolysis activity of wildtype (WT) *Re*LEH must be inhibited, which affords mainly the respective 1,2-diol and about 10% of the 6-membered Baldwin cyclization product. Here we present a rational mutagenesis-based design of *Re*LEH that selectively catalyzes the transformation of a range of epoxides and oxetanes to their cyclization-products, in which the distal HO- and RHN-groups serve as nucleophiles (Fig. 1b).

In this work, the genetically modified enzymes described herein perform the enantioselective synthesis of structurally diverse analogs of heterocycles accessible in fewer steps and in an ecologically and economically viable manner, while in conventional chemistry such syntheses are often lengthy and generally require the use of protecting groups and/or transition metals.

## Result and discussion

### Engineering *Re*LEH towards Baldwin cyclizations

Figure 1a displays the conventional catalytic mechanism of LEH, residues Y53 and N55 positioning water for nucleophilic attack, while D132 assists nucleophilic substitution and D101 ensures stabilization of the incipient negatively charged *O*-atom of the epoxy function[31]. We set out to disrupt water activation by appropriate mutagenesis, while keeping D101. Consequently, our focus was on Y53 and N55, but not on D101. In order to reprogram the enzyme catalytic properties, hydrophobic amino acids (Gly, Ala, Val, Pro, Leu, Ile, and Phe) in addition to WT were employed as building blocks at the residues Y53 and N55 with the aim to break the hydrogen bonding network (Fig. 2a). Degenerate codons GBA, WTK and Pro were rationally employed for library construction (Supplementary Figs. 2 and 3, and Supplementary Tables 1 and 2). To ensure 95% library coverage, ~180 clones were then screened in cell lysates toward the transformation of 5 mM *rac*-1 (Fig. 2b). Several hits with essentially eliminated WT hydrolytic activity were identified, including variants SZ532

**Fig. 3 | Substrate scope of Baldwin reactions catalyzed by LEH mutants.** Reaction conditions: LEH (0.1 g/mL wet cell), Substrates (5 mM), PBK buffer (50 mM, pH 7.4), Lysozyme (1 mg/mL), DNase I (6 U/mL), 0.5 mL total volume, 30 °C, 1000 rpm, 24 h.

(Y53L/N55P), SZ534 (Y53V/N55P), SZ609 (Y53I/N55A) and SZ611 (Y53F/N55A). Indeed, they enable Baldwin-type cyclization with little formation of undesired 1,2-diols (Fig. 2c). In the case of the double mutant SZ611 (Y53F/N55A), the formation of Baldwin product **2** is improved from 10% (WT) up to 78% (**2:3** = 7.8) (Fig. 2d). In parallel, to double check the drop-off of hydrolytic property, the WT enzyme and variants were also employed in the transformation of cyclohexene oxide, which lacks a hydroxyl group for possible intramolecular processes. Compared to WT, the hydrolysis conversion is largely decreased from 99 to 46% (Supplementary Table 3). These results suggest that the substitutions Y53F and N55A of *Re*LEH can effectively break the water binding network, thereby endowing Baldwin cyclization reactivity. This was our primary goal, to be followed by engineering notable regioselectivity (Baldwin vs. anti-Baldwin) and high enantioselectivity.

In order to exert greater evolutionary pressure, a second round of genetic manipulation based on rational saturation mutagenesis using mutant SZ611 (Y53F/N55A) as template was performed at the key active site residues Ile80, Val83, Leu114, and Ile116 surrounding the catalytic center D101 (Fig. 2a). We were guided by the enzyme mechanism and realization that hydrophobic amino acids should be chosen for exchange events at these hotspots. Moreover, based on earlier mutagenesis studies of LEH, QM cluster techniques had supported the mechanistic role of D101[32]. In the present study, we chose the triple code Val-Phe-Tyr which encodes hydrophobic amino acids, the double mutant Y53F/N55A serving as the template. After screening 576 transformants with automated gas chromatography (GC) for 95% library coverage, a distinctly improved mutant SZ616 (Y53F/N55A/I116V) was obtained, enabling a **2:3** ratio of 6.7 favoring the Baldwin reaction mode, although conversion to the anti-Baldwin product **3** was only 13%, the rest of complete conversion being **2**. Stereoselectivity varied, reaching a maximum of e.r. = 90:10. In curiosity-driven experiments, the mutants SZ612 (Y53F/N55A/I80F/L114V/I116V) and SZ621 (Y53F/N55A/I80F/I116V) produced only the Baldwin product **2** with none of **3** nor of the vicinal diol **4** as side-products (Fig. 2e, Supplementary Fig. 4).

**Expansion of substrate scope of *Re*LEH variants following Baldwin cyclization reactivity**
In order to evaluate the substrate scope of the engineered *Re*LEH mutants for the synthesis of heterocycles, a series of epoxy alcohols were prepared and tested. Most of the respective Baldwin products were obtained with good to excellent conversions and high cyclization mode selectivity without the need for additional mutagenesis, while diastereo- and enantioselectivity varied (Fig. 3). For example, variant SZ616 (Y53F/N55A/I80F/I116V) catalyzes the cyclization of the 5,6-epoxy alcohol **5** with sole formation of the Baldwin tetrahydropyran product **6** (99% conversion), and the diastereomeric ratio is 86:14.

**Selective desymmetrization reactions of oxetanes**
Natural enzymes that catalyze oxetane-opening reactions have rarely been reported[33,34]. We were all the more interested to learn how our engineered LEH mutants would perform using such substrates (Fig. 4). Initially, we explored the intramolecular desymmetrization reaction of substrate **9**, WT not showing any reaction. It is of note that WT still maintains hydrolysis ability towards cyclohexene oxide when simultaneously adding substrate **9** in the reaction system, suggesting that substrate **9** does not function as an inhibitor of WT *Re*LEH (Supplementary Fig. 5). We were pleased to discover that when using triple mutant SZ616 (Y53F/N55A/I116V), the tetrahydrofuran product **10** was obtained in 96% conversion with a 72:28 enantioselectivity ratio (e.r.). Upon testing variant SZ621 (Y53F/N55A/I80F/I116V) having the additional I80F mutation, stereoselectivity of the target product **10** was improved to 87:13 e.r. (Fig. 4, Supplementary Table 4). All other variants have no increased catalytic activity in this reaction. We therefore employed variants SZ616 and SZ621 as catalysts in the reaction of a range of structurally different oxetanes. Substrate scope and selectivities proved to be good to excellent in numerous cases (Fig. 4). A series of oxetanes bearing electron-donating and electron-withdrawing groups on the phenyl rings also underwent cyclization to afford the respective products (**10**, **12–32**) in high conversion with moderate stereoselectivity. Notably, substrates bearing dimethyl-substituted phenyl (**15**) and bicyclic naphthalene rings (**23**) were well tolerated under standard conditions, affording the cyclization products in conversion of 85% and 95%, respectively. Oxetanes **33**, **35**, **37**, and **39** with various alkyl groups or hydrogen atom, were accepted as substrates, and the corresponding products (**34**, **36**, **38**, and **40**) were obtained with 20–96% conversion and up to 96:4 e.r. In addition, even substrates **41** and **43** reacted to give chiral tetrahydropyran products. To demonstrate the synthetic practicality of this methodology, several preparative up-scaled reactions were performed for oxetanes **9** and **11**-**35**, employing the best mutant SZ616 at 50 mM substrate loading, resulting in moderate to good yields (53–88%) with moderate to excellent stereoselectivity.

Due to the widespread presence of *N*-heterocyclic compounds in natural products and pharmaceuticals, we extended our synthetic targets to obtain such compounds using aforementioned mutants (Fig. 5). This prompted us to examine the intramolecular opening of oxetanes with *N*-centered nucleophiles. To our delight, a wide variety of oxetanes underwent cyclization-opening with very high enantioselectivity and conversion. Change of substituents on the amide moiety was well tolerated (substrates **45**, **47–57**). Noteworthy is that compounds bearing *N*-aromatic substituents with electron-donating or withdrawing groups were readily converted into pyrrolidines (products **54**, **56**, and **58**) in high conversion with excellent enantioselectivity. Further examples of wide structural variation are shown in Fig. 5. Furthermore, cyclizations into the next larger ring to provide chiral piperidines with up to 98% conversion and an enantiomeric ratio of

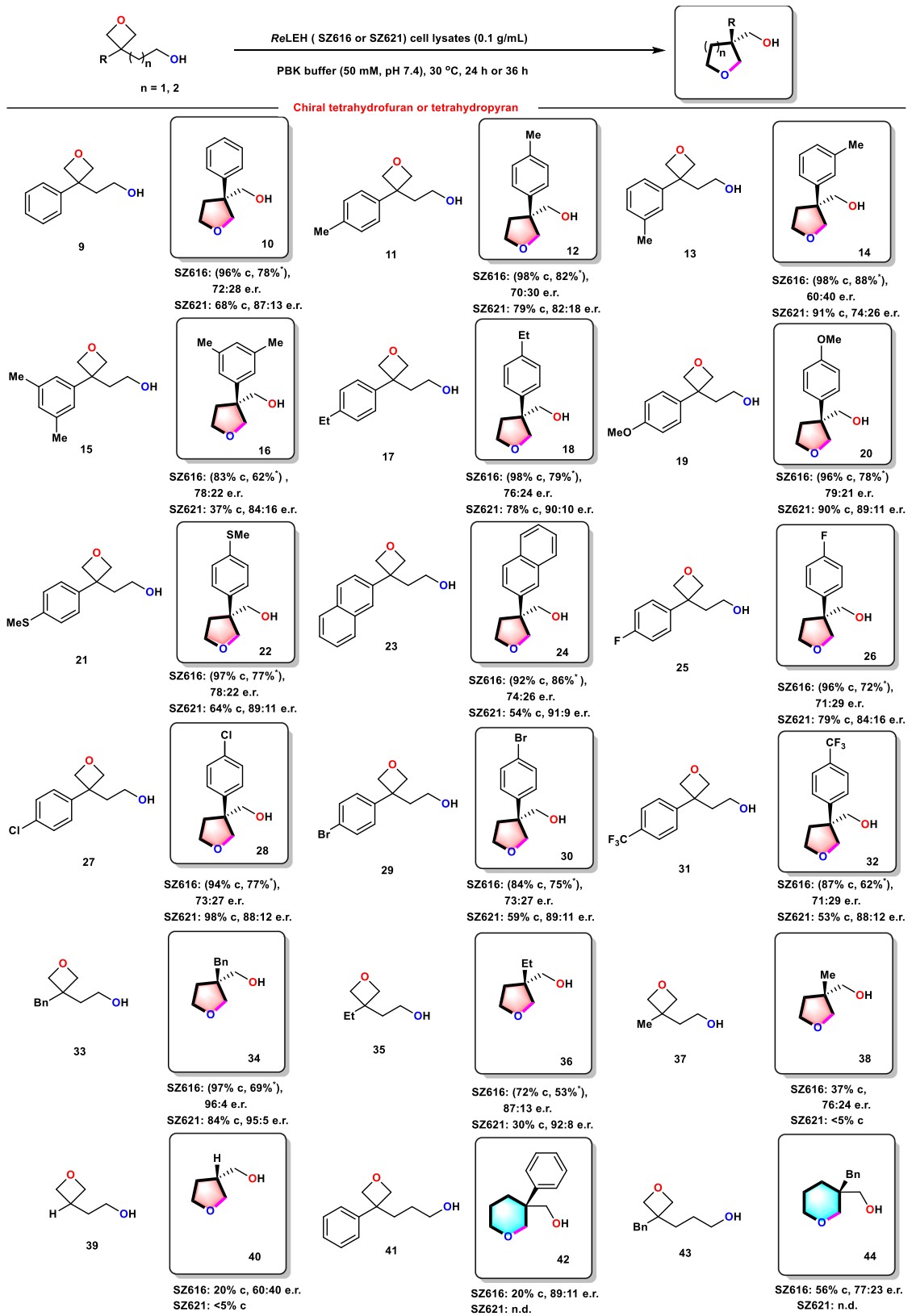

**Fig. 4 | Substrate scopes of *Re*LEH mutants in the synthesis of chiral tetrahydrofurans and tetrahydropyrans.** Reaction conditions: *Re*LEH (0.1 g/mL wet cell), Substrates (5 mM), PBK buffer (50 mM, pH 7.4), Lysozyme (1 mg/mL), DNase I (6 U/mL), 0.5 mL total volume, 30 °C, 1000 rpm, 24 h. *: Isolated yields of products with a substrate loading of 50 mM in 20 mL volume. c: Conversion (%). n.d.: Not detected.

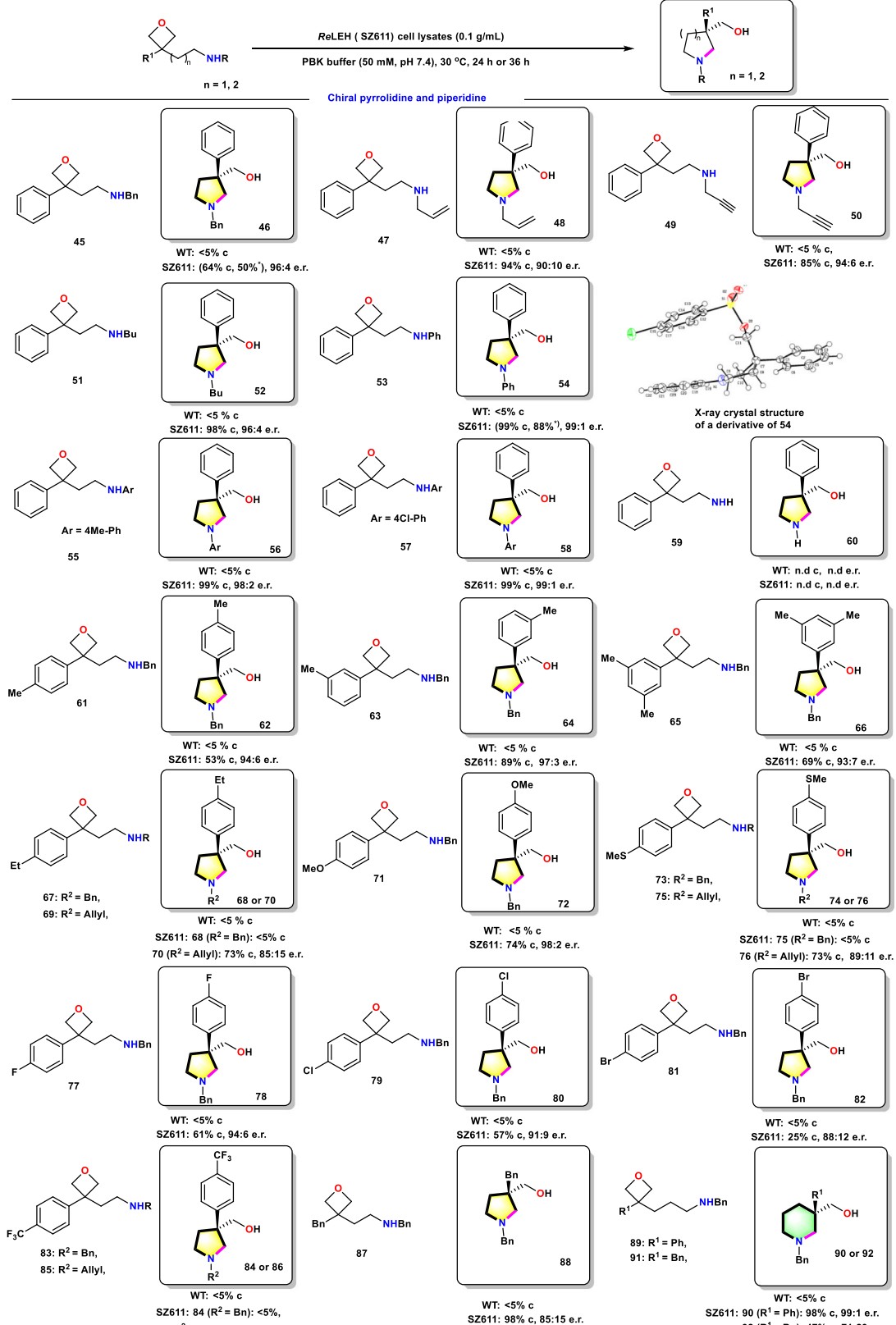

**Fig. 5 | Substrate scope of *Re*LEH variants in the synthesis of chiral pyrrolidines.**
Reaction conditions: *Re*LEH (0.1 g/mL wet cell), substrate (5 mM), PBK buffer
(50 mM, pH 7.4), Lysozyme (1 mg/mL), DNase I (6 U/mL), 0.5 mL total volume,
30 °C, 1000 rpm, 24 h. *: Isolated yields of products with a substrate loading of
50 mM in 20 mL volume. c: Conversion (%). n.d.: Not detected.

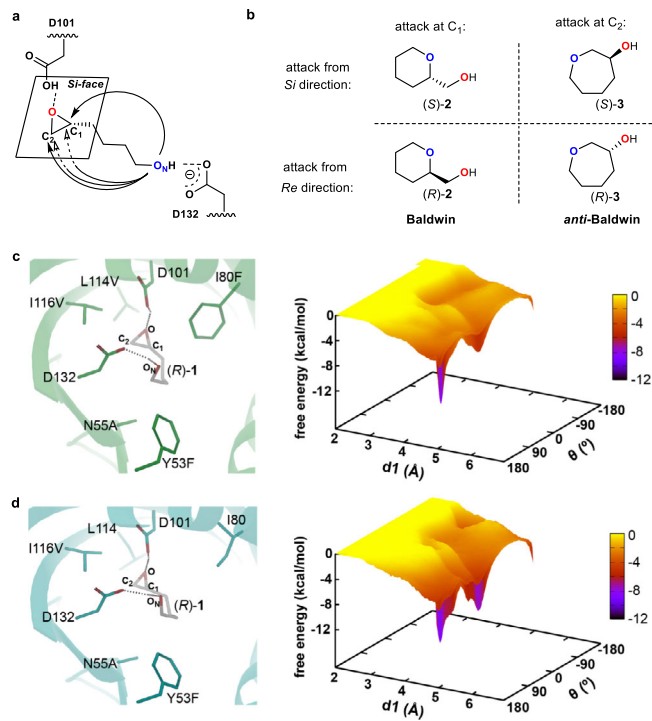

**Fig. 6 | Computational analysis of the designed mutants SZ612 and SZ616.**
**a** Proposed mechanism of LEH-catalyzed cyclization of the model substrate (R)-**1**.
**b** Schematic representation of the four possible products. **c** Model SZ612–(R)-**1** and
its free energy landscapes (kcal/mol) as a function of the dihedral angle θ (O$_N$-C1-
C2-O). **d** Model SZ616–(R)-**1** and its free energy landscapes (kcal/mol) as a function
of the dihedral angle θ (O$_N$-C1-C2-O).

99:1 were achieved, thereby establishing a simple method to synthe-
size chiral piperidine drug candidates. The absolute configuration of
the product was determined by X-ray crystal structure of a derivative
of **54** (Supplementary Table 5). In further work using purely synthetic
reagents, many of the *N*-heterocyclic compounds presented herein
were transformed by chemical means into other derivatives of
potential application in the production of chiral pharmaceuticals
(Supplementary Fig. 6). Intriguingly, the enzymatic product **46** can be
utilized to synthesize the natural product (R)-Coerulescine that has
analgesic effects as reported[35].

## Shedding light on the catalytic mechanism of Baldwin ring closure and desymmetrization

In order to gain insights on the active substrate-LEH interactions, the
X-ray structures of the key variants SZ612 and SZ616 were analyzed
(Supplementary Table 6). Consequently, the representative substrates
in complex with LEH mutants, including the models SZ612–(R)-**1** and
SZ616–(R)-**1**, were constructed by molecular docking. Following the
docking analysis, the epoxy ring *O*-atom of the substrate consistently
establishes a hydrogen-bond interaction with the protonated cataly-
tically active D101; the aforementioned hydrogen-bond network with
water molecular is terminated by substitutions Y53F and N55A,
enabling the substrate distal hydroxyl moiety to form a hydrogen-
bond interaction with the deprotonated D132 (Fig. 6a).

In the scenario of evolved Baldwin/anti-Baldwin selection and
stereoselectivity, two regioselective products are possible via Baldwin
or anti-Baldwin selection, each in two enantiomeric forms. Thus, four
different products can be formed (Fig. 6b). In detail, if the distal
deprotonated *O*-atom (dubbed O$_N$) attacks the reactive carbon (C1 or
C2) from the *Si*-face, the enantiomers (S)-**2** or (S)-**3** are generated. In
contract, the enantiomers (R)-**2** or (R)-**3** are formed when attack occurs
from the *Re*-face (Fig. 6b). In order to probe the conformational bias

for the substrate (R)-**1** in variants SZ612 and SZ616, two criteria were
employed to analyze the MD trajectories of models SZ612–(R)-**1** and
SZ616–(R)-**1**: (I) The (S)- or (R)-selectivity caused when the O$_N$ attack
occurs from the *Si*- or *Re*-face, by estimating the dihedral angle θ
between the four atoms O$_N$-C1-C2-O. If $0° < θ < 180°$, then O$_N$ attacks
from the *Si*-face, generating (S)-configured products; or if $−180° <
θ < 0°$, then O$_N$ attacks from the *Re*-face, to form (R)-configured
products. (II) The Baldwin or anti-Baldwin selectivity caused when the
O$_N$ attack occurs at C1 or C2, respectively, by determining the differ-
ence in attacking distances d1 (from O$_N$ to C1) and d2 (from O$_N$ to C2). If
d1 < d2, then C1 is attacked preferentially, yielding Baldwin products. If
d1 > d2, then C2 will be attacked, leading to anti-Baldwin products. As
such, 500-ns adaptively biased molecular dynamics (MD) simulations
were performed to investigate the free energy landscapes using the
dihedral angle θ and the distance d1 as collective variables. In model
SZ612–(R)-**1**, the dihedral angle θ was mainly gathered between 0° and
180° during the MD simulations (Fig. 6c), suggesting that attack from
the *Re*-face is unsupportive for SZ612. In contrast, the model SZ616–
(R)-**1** unveiled two peaks in terms of the dihedral angle θ (Fig. 6d),
revealing that the attack from *Si*- or *Re*-face has no preference in the
case of SZ616. Moreover, for both of the SZ612 and SZ616 mutants, the
slightly shorter distance d1 was observed during the simulations
(Supplementary Figs. 7a and 7b), indicating that the O$_N$ atom pre-
ferentially attacks C1. Furthermore, to better illustrate the bond
forming and breaking in the Baldwin/anti-Baldwin selection pathways
(Supplementary Movies 1–4), the hybrid QM/MM scheme was
employed, enabling the calculation the activation energy barriers. It is
of interest to note that the shorter distance d2 allows a relatively low
activation energy (Supplementary Figs. 7c and 7d), and hence the
formation of Baldwin product is favored for the two variants. These
observations are also in good agreement with the experiment results.

For the desymmetric oxetane reactions, we propose a putative
catalytic pathway, which mainly includes the following crucial pro-
cesses: (1) deprotonation of the amine group; (2) nucleophilic attack of
carbon atom (C2) of the oxetane; (3) protonation of oxygen atom of
oxetane (Fig. 7a). To further elucidate the mechanism, crystallographic
experiments were performed. We obtained the complex crystals of
SZ611-substrate **59** and SZ611-product **46** using the soaking methods
as described in **Material and Methods**. The complex structures of
SZ611-substrate **59** (PDB code: 7XEE) and SZ611-product **46** (PDB code:
7XEF) were determined to 1.88 Å and 1.82 Å, respectively (Supple-
mentary Table 6). SZ611 assembles into a homodimer structure with
four α-helices and five-strand β-sheets in each monomer, forming a
cavity that serves as a deep substrate-binding pocket, in which a sub-
stituent of substrate **59** or of product **46** can occur without steric
repulsion (Fig. 7b). The electron density map of such ligands and
surrounding key residues were defined with good quality (Fig. 7c, d). A
major portion of substrate **59** was clearly defined in the electron
density map, apart from a minor portion of the phenyl group. In
contrast, product **46** was well-defined in the electron density map,
including the phenyl group corresponding to that of substrate **59**. The
phenyl group is located in the hydrophobic local space formed by the
hydrophobic residues of L71, L74 and F75, which were clearly defined
in the electron density map. Two other essential mutations, Y53F and
N55A, have well-defined electron density maps that together expand
the hydrophobic local space, thereby leading to the preference of
intramolecular cyclization reactions. The molecular architecture of
substrate **46** in the binding pocket was mainly attributed to its multiple
interactions with catalytic residues D101 and D132 (Fig. 7e). The *N*-atom
of substrate **59** is hydrogen-bonded to D132, which commonly func-
tioned as a proton receptor for the deprotonation of an amine group,
or for deprotonation of water molecules in WT LEH and other LEH
mutants. The *O*-atom of oxetane is within the hydrogen-bonding dis-
tance to D101, which acts as a proton donor for the protonation of the
ring-opening *O*-atom, in agreement with an earlier study featuring the

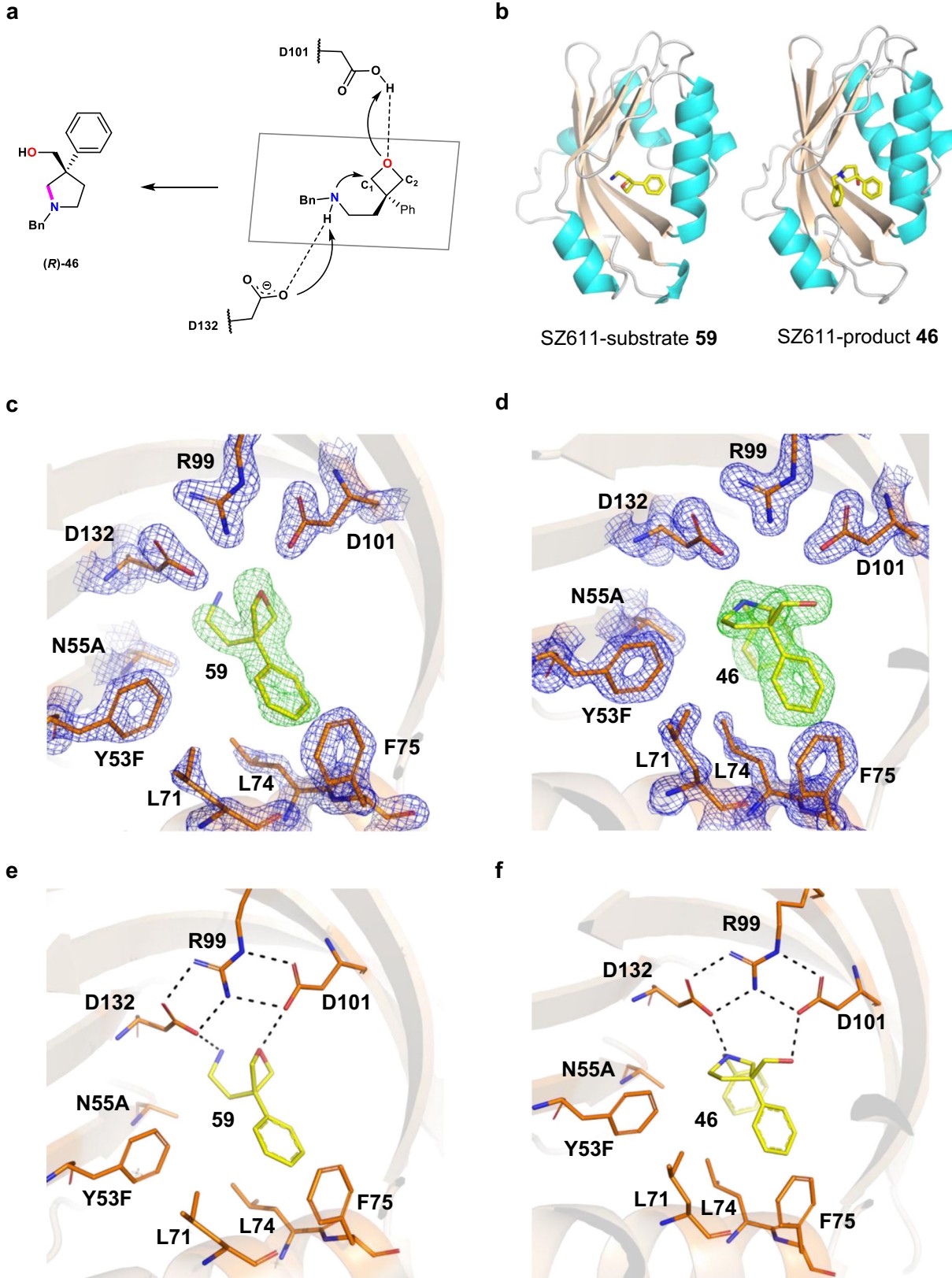

**Fig. 7 | Structural analysis of the designed mutant SZ611. a** Proposed mechanism of LEH-catalyzed selective desymmetrization reactions of oxetanes. **b** Overall structure of LEH bound to substrate **59** and product **46** (yellow). **c** Electron density for substrate **59**. **d** Electron density for substrate product **46**, Fo-Fc map contoured at 3.0σ, green mesh and protein side chains shown in blue mesh, 2Fo-Fc map contoured at 2.0σ. **e** Hydrogen-bonding interactions between substrate **59** and protein side chains. **f** Hydrogen-bonding interactions between product **46** and protein side chains. Dash: hydrogen-bonding interactions.

X-ray structure of WT LEH[36]. Product **46** shares a similar conformation with substrate **59** considering the hydrogen-bonding networks of amines and hydroxyl groups with D132 and D101 (Fig. 7f), which together predicts a pathway for the LEH-catalyzed selective desymmetrization reactions of oxetanes. In the catalytic reactions, residue D132 acts as a proton acceptor to promote the deprotonation of the amino group. The deprotonated *N*-atom then nucleophilically attacks the oxetane moiety, resulting in ring opening, followed by the protonation of the *O*-atom of the oxetane precursor with D101 as the proton donor.

We have rationally engineered the epoxide hydrolase *Re*LEH with construction of a general platform for Baldwin cyclization reactivity of hydroxy- and amino-epoxides and oxetanes. The genetic reprogramming of the enzyme enables an efficient biocatalytic route to chiral *N*- and *O*-heterocycles of considerable significance in the pharmaceutical field. Structurally diverse chiral heterocycles can now be accessed from appropriate epoxides and even oxetanes with high reactivity and selectivity. Considering the fundamental roles of chiral heterocycles in the biological world and the pharmaceutical industry, we anticipate that these fully genetically encoded enzymes will provide a starting point to further extend the toolbox of ring-expansion catalysts. The rational enzyme design strategy presented in this study also provides a guidance for engineering other natural enzymes with the aim to access hitherto never observed functions.

## Methods

### General
NMR spectra were recorded on Bruker AV 400 MHz instrument (400 MHz for $^1$H, 100 MHz for $^{13}$C, and 376 MHz for $^{19}$F). Chemical shifts were reported in ppm down field from internal Me$_4$Si and external CCl$_3$F, respectively. Multiplicity was indicated as follows: s (singlet), d (doublet), t (triplet), q (quartet), m (multiplet), dd (doublet of doublet), br (broad). A waters G2-XS qtof instrument was used to generate the high-resolution mass spectrometry (HRMS) spectra data. Small molecular X-ray structural analysis was carried out using Bruker APEX-II CCD instrument. SHIMADZU 2030 HPLC and GC systems were used for HPLC and GC analysis, respectively. KOD Hot Start DNA Polymerase (Novagen) and restriction enzyme Dpn I (NEB) were used in this work. The oligonucleotides were synthesized by Life Technologies. Plasmid preparation kit was ordered from Zymo Research, and PCR gel extraction kit was bought from QIAGEN. DNA sequencing was conducted by GATC Biotech. Tetrahydrofuran (THF), diethyl ether, and toluene were distilled from sodium/benzophenone prior to use; CH$_2$Cl$_2$ was distilled from CaH$_2$; All purchased reagents were used without further purification. Analytical thin layer chromatography (TLC) was performed on 0.20 mm silica gel plates, while silica gel (200–300 mesh) was used for flash chromatography. All the silica gels were purchased from Qingdao Haiyang Chem. Company, Ltd.

### PCR-based methods for library construction
KOD Hot Start polymerase was used to construct mutant libraries via the megaprimer approach. 30 μL water, 5 μL KOD hot start polymerase buffer (10×), 3 μL 25 mM MgSO$_4$, 5 μL 2 mM dNTPs, 2.5 μL DMSO, 0.5 μL (50-100 ng) template DNA, 100 μM primers mix 0.5 μL each and 1 μL KOD hot start polymerase were added as a 50 μL reaction mixtures. The PCR cycles used for generating short fragment: 95 °C 3 min, (95 °C 30 s, 56 °C 30 s, 68 °C 40 s) × 32 cycles, 68 °C 120 s, 16 °C 30 min. While in mega-PCR: 95 °C 3 min, (95 °C 30 s, 60 °C 30 s, 68 °C 5 min 30 s) × 24 cycles, 68 °C 10 min, 16 °C 30 min. The PCR products were analyzed on agarose gel by electrophoresis. The digestion was carried out at 37 °C for more than 3 h by adding 2 μL NEB CutSmart™ Buffer and 2 μL Dpn I in 50 μL PCR reaction mixture. After digestion, 1 μL PCR products were directly transformed into electro-competent *E. coli* BL21(DE3) to create the final library for Quick Quality Control and screening. Primers used in this work are listed in Supplementary Tables 1 and 2.

### Screening procedures
Colonies were picked and transferred into deep-well plates, which contain 300 μL LB medium with 50 μg/ml carbenicillin in each well. After culturing overnight at 37 °C with shaking, 120 μL was transferred to glycerol stock plate and stored at −80 °C. Then, 800 μL TB medium with 0.5% (m/v) lactose and 50 μg/mL carbenicillin was added directly to the culture plate, shaking at 28 °C for protein expression. After 8 h, the cell pellets were harvested and washed using 400 μL 50 mM pH 7.4 potassium phosphate buffer and centrifuged for 10 min 1100 × *g* at 4 °C. Then, the pellets were resuspended in 400 μL of the same buffer with 6 U DNase I and 1 mg/mL lysozyme for breaking the cell at 30 °C for 1 h with shaking. The crude lysate was centrifuged at 1100 × *g* and 4 °C for 30 min. Afterwards, 300 μL supernatant of the lysate was transferred into new deep-well plates, which preloaded 100 μL 5 mM substrate *rac*-**1** and 5% acetonitrile as co-solvent. After reaction at 30 °C and 800 rpm for 14-16 h, the product and remaining substrate were extracted using equal volumes of ethyl acetate (EtOAc) for GC analysis by chiral column Hydrodex-*β*-TBDAc, 25 m × 0.25 mm ID as the follows condition: 110 °C, 5 °C/min, 135 °C, 20 °C/min, 220 °C hold 2 min. N$_2$: 1.5 bar.

### X-ray structure determination
The proteins, SZ611 at 10 mg/mL, SZ612 at 10 mg/mL and SZ616 at 20 mg/mL, containing solution (25 mM HEPES pH 7.5, 150 mM NaCl) were mixed in a 1:1 ratio with the reservoir solution in a final volume of 2 μL and equilibrated with the reservoir solution. The SZ611 crystals were grown in 0.2 M NaCl, 0.1 M Bis-Tris pH 5.5, 30% (w/v) PEG 3350, and supplemented with additives (0.5% (w/v) casein, 0.5% (w/v) hemoglobin, 0.005% (w/v) pepsin, 0.005% (w/v) protease, 0.005% (w/v) proteinase K, 0.005% (w/v) trypsin, and 0.02 M HEPES sodium pH 6.8) at a ratio of 10:1, using the sitting-drop vapor diffusion method at 20 °C. The complex crystals of SZ611-substrate **59** and SZ611-product **46** were made by soaking SZ611 crystals in growth buffer containing 32 mM substrate **59** and substrate **45** for 10 min, respectively. The SZ612 crystals were grown in 0.2 M ammonium acetate, 25% (w/v) PEG 3350, and the SZ616 crystals were grown in 0.2 M ammonium formate, 15% (w/v) PEG 3350, both of which used the sitting-drop vapor diffusion method at 4 °C. All crystals were mounted in a cryoloop and frozen in liquid nitrogen after immersion in a well solution containing 10–25% glycerol. All data were collected at 100 K. Diffraction data of the SZ611 and SZ612 were measured at the wavelength of 1.03312 Å and SZ616 at 0.97853 Å at BL17B1 beamline of Shanghai Synchrotron Radiation Facility. Diffraction data of both SZ611-substrate **59** and SZ611-product **46** were measured at the wavelength of 1.5418 Å on an R-AXIS IV$^{++}$ imaging plate detector. All data sets were processed by using the HKL-2000 package[37]. The crystal structures were solved by molecular replacement method using the program PHASER (version 2.7.16)[38] using the structure of wild-type LEH (PDB code 1NWW) as a search model. Further refinement was performed with PHENIX (1.11.1_2575)[39], and the models were rebuilt manually with Coot (0.9.8.1)[40]. Data collection and refinement statistics were summarized in Supplementary Table 6.

### Model preparation and model generation and substrate docking
Schrödinger Maestro software was used to prepare protein and ligand structures[41]. The X-ray structures of variants solved in this study (PDB IDs 7VWD, 7VX2, and 7VWM) were used as templates. The substrate (*R*)-**1** was separately docked into 7VX2 and 7VWM and generated the models SZ612–(*R*)-**1** and SZ616–(*R*)-**1**. The docking of substrate to the substrate binding pocket was carried out by Glide[42] and Induced fit docking (IFD)[43].

### MD simulations and QM/MM calculations
All of the MD runs were performed by using the GPU version of AMBER 2016[44]. *Apo*-proteins was protonated at pH 7.4 (to be consistent with

the experimental conditions) by H++[45]. The Amber ff14SB force field[46] and the TIP3P water model[47] were employed, and the solvent layer was set as 10 Å in MD runs. The parameters and charges for substrates (*R*)-**1** were generated with the antechamber module using the AM1-BCC charge model and the general AMBER force field (GAFF)[48]. After proper parameterizations and setup, the resultant system's geometries were minimized in two-stage, the first stage only focused on the positions of solvent molecules and ions, and the second stage is an unrestrained minimization of all the atoms in the system. And then the system was heated from 0 to 303 K under the NVT ensemble for 50 ps. Subsequently, the systems were maintained for 50 ps of NPT equilibration at 303 K and 1.0 atm, using Langevin-thermostat (ntt=3) with a collision frequency of 2 ps$^{-1}$ and pressure relaxation time of 1 ps. A weak restraint of 10 kcal mol$^{-1}$ Å$^{-2}$ was performed on the protein residues during heat and density equilibrations. The systems were further equilibrated for 1 ns without restraints. For models SZ612–(*R*)-**1** and SZ616–(*R*)-**1**, the adapted biased 500-ns MD simulations were carried out, the free energy landscapes were calculated at the temperature 1200 K, following the procedures reported in the previous work[49]. During all MD simulations, the covalent bonds containing hydrogen were constrained using SHAKE algorithm[50]. The hybrid QM/MM calculations were performed by using GTKDynamo tool (version 1.9.0)[51], where QM part is treated using the PM6 semiempirical method[52] and the MM part using the AMBER force field. Residues 53, 55, 80, 99, 101, 114, 116, and 132 along with the substrate were chosen as QM region[53]. The catalytic processes of the two models SZ612–(*R*)-**1** and SZ616–(*R*)-**1** to generate Baldwin and anti-Baldwin products were visualized in Supplementary Movies 1–4.

### Reporting summary

Further information on research design is available in the Nature Portfolio Reporting Summary linked to this article.

## Data availability

The authors declare that all the data supporting the findings of this study are available within the main text, and Supplementary Information documents. The X-ray structural data have been deposited in the Protein Data Bank (www.rcsb.org) with accession codes 7VWD, 7VX2, 7VWM, 7XEE and 7XEF respectively. Crystallographic data for the structures reported in this Article have been deposited at the Cambridge Crystallographic Data Centre, under deposition numbers: 2162958 (Derivative of the product **54**). This data can be obtained free of charge from The Cambridge Crystallographic Data Center via www.ccdc.cam.ac.uk/data_request/cif. Additional data supporting the findings of this study are available from the corresponding author on request.

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

## Acknowledgements

Dedicated to the 10th anniversary of Tianjin Institute of Industrial Biotechnology, CAS. This work was supported by the National Key Research and Development Program of China (No. 2019YFA0905100), the National Natural Science Foundation of China (Nos. 32171474, 21961142015, and 92156025), the Natural Science Foundation of Tianjin (No. 21JCJQJC00110), the Biological Resources Programme, CAS (No. KFJ-BRP-009) and Tianjin Synthetic Biotechnology Innovation Capacity Improvement Project (TSBICIP-CXRC-009). GQ thanks Youth Innovation Promotion Association, CAS (No. 2021175) for additional financial support. MTR thanks the Max-Planck-Society for support. The authors would like to thank Xu Han and Weidong Liu for helping in the crystallographic data collection.

## Author contributions

Z.S., M.T.R., J.-A.M., and G.Q. conceived and directed the project. J.-K.L., G.Q., and X.L. designed and performed the experiments. Y.T., C.C., F.Z., and W.Z. assisted the substrate synthesis and interpreted the data. G.Q., Z.S., M.T.R., J.-A.M., J.-K.L., and X. L. wrote and revised the manuscript.

## Competing interests

Parts of this work are included in patent applications by the Tianjin Institute of Industrial Biotechnology (TIB), Chinese Academy of Sciences, covering LEH mutants and related methods for their catalytic synthesis of chiral oxygen/nitrogen heterocyclic compounds (application number: CN202111491732.1; Z.S., J.-K.L., G.Q., and W.Z.). The patent has been submitted through TIB. The remaining authors declare no other competing interests.
