## [Peer Review File · Nature Communications]

Rational enzyme design for enabling biocatalytic Baldwin cyclization and asymmetric synthesis of chiral heterocyclesREVIEWER COMMENTS

Reviewer #1 (Remarks to the Author):

The submitted manuscript from Li et al. reports about the design of selected epoxide hydrolase (LEH) variants aimed at the preparation of chiral N- and O-heterocycles.

The paper is very interesting and original. Several new products with high applicative potential could be obtained by exploiting a relatively simple engineering strategy. However, some points are not sufficiently clear in my opinion.

Therefore, I suggest the Editor to accept it after minor revisions.

Specific comments:

- p. 5, lines 6 and 18: Pivotal work published by Faber's group on the biocatalytic EHS-catalyzed hydrolysis/cyclization cascades of methylene-interrupted bis-epoxides, e.g., Chem. Eur. J. 2004, 10, 3467 – 3478, DOI: 10.1002/chem.200400061, and following ChemBioChem 2009, 10, 1697 – 1704, DOI: 10.1002/cbic.200900176, deserves to be cited and discussed in the Introduction.
- p. 5, line 11: the sentence "it is highly sensitive to the hydrolysis of the epoxide substrates with the formation of the respective chiral vicinal diols" is not very clear to me, specifically the use of the word "sensitive" seems not much appropriate in this context, please rephrase.
- p. 6, lines 5-7: about the sentences: "... residues Y53, N55 and D132 positioning and activating water for nucleophilic attack, while D101 ensures stabilization of the incipient negatively charged O-atom of the epoxy function. We set out to disrupt water activation by appropriate mutagenesis, while keeping D101. Consequently, our focus was on Y53 and N55", it is not completely clear why D132 was not considered as a possible target in variants design. Please explain and/or rephrase.
- p. 6, line 12, and following text: the extent of suppression of the hydrolytic activity of LEH variants varies in a consistent way going from substrate rac-1 (full suppression) to cyclohexene oxide (from 99% conv. to 46%, Fig. S4 in SI). How do the authors explain these differences? Is the suppression of the hydrolytic activity "easier" with rac-1 because it is a worse substrate for LEH (total conversion is only 14% in Fig. 2c)?
- p. 9, line 9: substrate 9 is not hydrolyzed by the wild-type LEH, but can be converted into 10 by different variants, thus demonstrating a good active site interaction. Is it maybe an inhibitor of wild-type LEH?
- p. 17, lines 11-14: again, I don't believe that it is totally correct to say that "Two other essential mutations, Y53F and N55A, have well-defined electron density maps that together expand the hydrophobic local space, thereby excluding water molecules from the catalytic pocket, which would otherwise lead to 1,2-diol formation", because variants with these two mutations are still pretty active epoxide hydrolases (see Fig. S4 in SI), therefore there is still room for water molecules in the active site, unless probably the case in which "bad" and bulky/hydrophobic substrates, e.g., those shown in Fig. 5, are used. Please rephrase accordingly.

Reviewer #3 (Remarks to the Author):

This is a nicely written paper about the biocatalytic synthesis of saturated oxygen and nitrogen heterocycles via epoxide and azetidine opening. The strategy is thoughtful, the scope is quite good, and there is some structural insight into how the reactions work. I think it should get published with minor revisions.

One complaint might be that the yields and enantioselectivities are modest by small molecule standards, but I actually think that a reaction that generates a diverse range of products with moderate efficiency is much more interesting than a reaction that generates a couple of closely related products with high efficiency. It is precisely the reactions that are most general that get used, and, with a biocatalytic platform in place, it is entirely feasible to engineer a specific variant of particular interest. A specialized unicorn for one substrate is a curiosity; a swiss army knife can become a workhorse. To that end, it would be nice to see some data for pharmaceutically-relevant

heterocycles. Of course, some of these will be nucleophilic and incompatible, but there might be some surprising scope given the enzymatic context.

I would also be interested to hear the authors comment on the relationship, if any, between yield and ee. Traditionally, enzymes have been thought of as having these highly specific pockets meant for one substrate. The thinking was that if you got any reactivity at all, it would be highly enantioselective. Now, with these engineered variants, I think we're seeing much more scope and this yield-ee relationship break down. To me, that's a very good thing, as it opens up the possibility that enzymes might become much more general tools.

As someone who has his roots in mechanistic analysis, I was intrigued to see the mechanistic analysis here. I've done some metadynamics before, but not adaptively biased MD. My understanding is that the idea is roughly the same—to artificially flatten out the free energy surface to improve sampling efficiency, and then undo the flattening to retrieve the surface itself. If that's correct, then are we effectively conducting a ground state analysis of the “near attack conformations”? One could imagine that a more rigorous analysis would involve using QM/MM methods to model the actual bond forming step. I think this kind of ground state reasoning is quite common in enzymology (and with good reason), but it might be a good idea to state this caveat. Furthermore, the sampling of different pathways and mechanisms is very limited here. I view this analysis at the level of an inspirational cartoon that may inspire rational choices in the design of future enzyme variant (or libraries), rather than a detailed parsing of how exactly the enzyme works. The latter would require far more experiments than are warranted here.

Overall, I commend the authors on a nice paper and look forward to seeing it in print.

Reviewer #4 (Remarks to the Author):

Overall, I believe these structures have not been modelled fully and would recommend the authors to revisit the structures and take more care to address the difference density that is present in the structures. I have made a list of the specific issues I have found within each structure.

Of particular concern are the very high B-factors of the ligand molecule in 7XEE which much of the paper hinges on. I re refined the ligands against the mtz provided and remade the ligand and restraints and the B-factors were much lower. There is also a possibility of an alternate conformation of the ligand based on the presence of difference density when a single ligand conformation was modelled (see pictures in attachment).

Structure 7VWD

Difference map peaks >5 sigma

- Number 2 and 6 look like a PED molecule not the waters that have been modelled
- Number 4 should have a coordinating water added
- Number 10 should have a water modelled

Chain C has clear density for residues before residue number 5, up to residue -2 can easily be modelled.

Glu138 in chains A, C and D are poorly modelled to the density.

Leu147 backbone is out of the density

The rotamer outliers for chain D are much higher compared to other chains, should be addressed.

Structure 7XEE

The B-factors for both the CXI ligands are very high, ~60 and ~100 in chains A and B, respectively. This is especially troubling as the B-factors of the surrounding residues are around 15. I would

suggest the authors re refine their ligand with a lower occupancy to address this problem. The ligands also have a very high Z score for both bond lengths and angles.

Again, a number of difference peaks at 5 sigma or above should be addressed, some look like potential metal binding sites based on the coordination surrounding the peaks.

Ser37 in chain B should have a second conformation modelled.

Ala12 is overlapping with a water molecule, this should be addressed.

Structure 7XEF

Incorrectly modelled side chains

- Arg9 – chain A
- Glu138 – chain C and D
- Leu147 – Chain C
- Ser111 – Chain D – Add alternative conf
- 121Glu – Chain D- Add alternate conf
- 78Met – Chain D – Add alternate conf
- 49Glu – Chain D – Add alternate conf

There is density in chain B for residues before number five up to -3 of the tag.

Structure 7VXZ

There is density in chain C and D for residues before number five up to -3 of the tag. Currently waters are filling the density.

There are a good number of difference density peaks around 5 sigma that should be addressed within the structure.

Arg99 in chain C has an incorrect rotamer, with the NE atom facing away from Asp101

There are unmodelled blobs present in the active site of chain C that should be addressed.

Ard99 in chain D is not interacting with Asp132 as I would expect.

Structure 7VMW

There are over 50 difference map peaks around 5 sigma which could and should be addressed.

Gln121 is modelled incorrectly

PEG 3 in chain G should be remodelled into the difference density present and possibly has Asp101 modelled with an alternate conformation

Met78 on chain D should have an alternate conformation added.

Density is present to model residue 4 in chain A.

Reviewer #1 (Remarks to the Author):

The submitted manuscript from Li et al. reports about the design of selected epoxide hydrolase (LEH) variants aimed at the preparation of chiral N- and O-heterocycles.

The paper is very interesting and original. Several new products with high applicative potential could be obtained by exploiting a relatively simple engineering strategy. However, some points are not sufficiently clear in my opinion. Therefore, I suggest the Editor to accept it after minor revisions.

Response: Thank you very much for the positive comments. The following is our point-by-point response to the specific comments.

Specific comments:

- p. 5, lines 6 and 18: Pivotal work published by Faber's group on the biocatalytic EHs-catalyzed hydrolysis/cyclization cascades of methylene-interrupted bis-epoxides, e.g., Chem. Eur. J. 2004, 10, 3467 – 3478, DOI: 10.1002/chem.200400061, and following ChemBioChem 2009, 10, 1697 – 1704, DOI: 10.1002/cbic.200900176, deserves to be cited and discussed in the Introduction.

Response: Thanks, the above two mentioned papers are excellent examples of hydrolysis of bis-epoxide, they have been cited in the revised manuscript.

- p. 5, line 11: the sentence “it is highly sensitive to the hydrolysis of the epoxide substrates with the formation of the respective chiral vicinal diols” is not very clear to me, specifically the use of the word “sensitive” seems not much appropriate in this context, please rephrase.

Response: We have changed the word “sensitive” to “selective”.

- p. 6, lines 5-7: about the sentences: “... residues Y53, N55 and D132 positioning and activating water for nucleophilic attack, while D101 ensures stabilization of the incipient negatively charged O-atom of the epoxy function. We set out to disrupt water activation by appropriate mutagenesis, while keeping D101. Consequently, our focus was on Y53 and N55”, it is not completely clear why D132 was not considered as a possible target in variants design. Please explain and/or rephrase.

Response: Thanks for the suggestion. The D132 residue formally deprotonates a molecule of water *de facto* providing a crucial assistance to the key step of nucleophilic substitution (please see ChemBioChem, 2020, 21(13):1868-1874), we therefore didn't choose it as a mutagenesis candidate. To make it clearer, we have rephrased the relative sentences in the revised manuscript.

- p. 6, line 12, and following text: the extent of suppression of the hydrolytic activity of LEH variants varies in a consistent way going from substrate rac-1 (full suppression) to cyclohexene oxide (from 99% conv. to 46%, Fig. S4 in SI). How do the authors explain these differences? Is the suppression of the hydrolytic activity “easier” with rac-1 because it is a worse substrate for LEH (total conversion is only 14% in Fig. 2c)?

Response: We believe that these differences are mainly due to the distinct catalytic abilities of wild-type LEH in hydrolyzing the two types of substrates rac-1 and cyclohexene oxide. As we reported in the previous literatures (please see: J. Am. Chem. Soc. 2010, 132, 44, 15744–15751. and Angew. Chem. Int. Ed. 2015, 54, 12410 –12415), the structure for cyclohexene oxide is closer to the natural substrate of LEH. Therefore, LEH is more active in the hydrolysis of cyclohexene oxide relative to rac-1. Even though the mutants disrupt the hydrogen bonding network of water molecules, resulting in a decrease in the hydrolysis activity of LEH towards cyclohexene oxide (from

99% conv. to 46%), some residual hydrolytic activity was retained, since this may be due to that that cyclohexene oxide is a “better substrate” for LEH, hence it is difficult to fully suppress the hydrolytic activity. On the other hand, rac-1 is a “worse substrate”, and therefore the hydrolytic activity towards it was fully suppressed when catalyzed by the mutants under the defined reaction conditions.

- p. 9, line 9: substrate 9 is not hydrolyzed by the wild-type LEH, but can be converted into 10 by different variants, thus demonstrating a good active site interaction. Is it maybe an inhibitor of wild-type LEH?

Response: Thanks, it is an interesting question. In order to test whether the undetectable activity toward substrate 9 is caused by inhibition, we have performed three experiments, I, II and III. All the three reaction systems are the same, except that in experiment I only substrate 9 was added, in II only the natural substrate cyclohexene oxide was used, while in the case of III simultaneously substrate 9 and cyclohexene oxide were added. If WT LEH shows no activity or relatively lower activity toward cyclohexene oxide in experiment III compared to that in experiment II, it could be an indication that substrate 9 acts as an inhibitor. Otherwise, if similar activity in both experiments II and III were to be observed, then it would support the notion that substrate 9 is simply not recognized by the enzyme rather than an inhibitor. As shown in the newly provided Fig. S6, the experiments II and III showed that WT LEH can completely transform cyclohexene oxide to the vicinal diol product, with or without the addition of substrate 9. Therefore, substrate 9 appears not to function as an inhibitor of WT LEH.

- p. 17, lines 11-14: again, I don't believe that it is totally correct to say that “Two other essential mutations, Y53F and N55A, have well-defined electron density maps that together expand the hydrophobic local space, thereby excluding water molecules from the catalytic pocket, which would otherwise lead to 1,2-diol formation”, because variants with these two mutations are still pretty active epoxide hydrolases (see Fig. S4 in SI), therefore there is still room for water molecules in the active site, unless probably the case in which “bad” and bulky/hydrophobic substrates, e.g., those shown in Fig. 5, are used. Please rephrase accordingly.

Response: We have rephrased the mentioned sentences accordingly in the revised manuscript. Thank you again!

Reviewer #3 (Remarks to the Author):

This is a nicely written paper about the biocatalytic synthesis of saturated oxygen and nitrogen heterocycles via epoxide and azetidine opening. The strategy is thoughtful, the scope is quite good, and there is some structural insight into how the reactions work. I think it should get published with minor revisions.

Response: Thank you very much for the positive comments.

One complaint might be that the yields and enantioselectivities are modest by small molecule standards, but I actually think that a reaction that generates a diverse range of products with moderate efficiency is much more interesting than a reaction that generates a couple of closely related products with high efficiency. It is precisely the reactions that are most general that get used, and, with a biocatalytic platform in place, it is entirely feasible to engineer a specific variant of particular interest. A specialized unicorn for one substrate is a curiosity; a swiss army knife can

become a workhorse. To that end, it would be nice to see some data for pharmaceutically-relevant heterocycles. Of course, some of these will be nucleophilic and incompatible, but there might be some surprising scope given the enzymatic context.

Response: Thanks for the interesting suggestion. This study provides a biocatalytic platform to synthesize diverse chiral heterocycles in a general manner. In addition, we conducted the experiments in the direction of producing pharmaceutically-relevant heterocycles. For instance, the product 46 was successfully converted to three distinct derivatives 94, 95 and 96 with a yield of 85%, 80% and 76%, respectively. Moreover, following the reviewer's suggestion, we conducted a new transformation: Product 65 was successfully utilized to synthesize the natural product (*R*)-Coerulescine, a pharmaceutical compound that has analgesic effect (please see Fig. S8).

I would also be interested to hear the authors comment on the relationship, if any, between yield and ee. Traditionally, enzymes have been thought of as having these highly specific pockets meant for one substrate. The thinking was that if you got any reactivity at all, it would be highly enantioselective. Now, with these engineered variants, I think we're seeing much more scope and this yield-ee relationship break down. To me, that's a very good thing, as it opens up the possibility that enzymes might become much more general tools.

Response: Thanks for the comment, this is actually a very interesting topic. When using enzymes as catalysts, it is intriguing to note that the yield and ee values do not necessarily correlate. Moreover, one of the recent charming discoveries in enzymology concerns the possibility that enzymes may exhibit catalytic promiscuity, catalyzing mechanistically distinct transformations or accepting structurally diverse substrates (please see the relative review on this topic: Nature 2022, 606, 49-58), which we have newly cited it as ref21. Although the promiscuous catalytic activities are typically very low, they indeed provide starting points for further improvement by protein engineering means.

As someone who has his roots in mechanistic analysis, I was intrigued to see the mechanistic analysis here. I've done some metadynamics before, but not adaptively biased MD. My understanding is that the idea is roughly the same—to artificially flatten out the free energy surface to improve sampling efficiency, and then undo the flattening to retrieve the surface itself. If that's correct, then are we effectively conducting a ground state analysis of the “near attack conformations”? One could imagine that a more rigorous analysis would involve using QM/MM methods to model the actual bond forming step. I think this kind of ground state reasoning is quite common in enzymology (and with good reason), but it might be a good idea to state this caveat. Furthermore, the sampling of different pathways and mechanisms is very limited here. I view this analysis at the level of an inspirational cartoon that may inspire rational choices in the design of future enzyme variant (or libraries), rather than a detailed parsing of how exactly the enzyme works. The latter would require far more experiments than are warranted here.

Response: Thanks again for the comments and suggestions. Likewise metadynamics, adaptively biased MD is mainly used to improve sampling efficiency, which is beneficial for the ground state analysis of near attack conformations (NACs). Based on the theory of NAC, the catalytic efficacy of the enzyme LEH is dependent on how often the nucleophile and electrophile are present in properly positioned poses. In our work, the distances d_1 (from O_N to C_1) and the dihedral angle θ ($O_N-C_1-C_2-O$) were adopted as two NAC parameters to evaluate the occurrence of selectivities

(Baldwin and anti-Baldwin regioselectivity, *R*- and *S*-enantioselectivity) during the MD simulations. More details provided in Section 2.4. On the other hand, the LEH catalytic mechanism and reaction pathways have been well-documented in previous QM/MM (Biochimica et Biophysica Acta 2012, 1824, 263-268), QM/MM MD (ACS Catal. 2018, 8, 5698-5707) as well as QM cluster-based (J. Am. Chem. Soc. 2018, 140, 310-318) studies. Consequently, we didn't put much effort into this scope. However, it is really a good suggestion that modelling the bond forming/breaking, which may be inspirational for readers. In the revision work, we therefore employed the hybrid QM/MM scheme to calculate the barriers for ring opening and closing steps involved in the cyclization and desymmetrization by two alternative Baldwin/anti-Baldwin pathways, where QM part is treated using the PM6 semiempirical method and the MM part using the Amber force field. The calculated activation energy barrier is shown in Fig. S9, while the catalytic processes and covalent bond changes were visualized in Movies S1-S4.

Movie S1. The bond forming and breaking in the SZ612-catalyzed transformation of substrate (*R*)-1 to Baldwin product (*S*)-2.

Movie S2. The bond forming and breaking in the SZ612-catalyzed transformation of substrate (*R*)-1 to anti-Baldwin product (*R*)-3.

Movie S3. The bond forming and breaking in the SZ616-catalyzed transformation of substrate (*R*)-1 to Baldwin product (*S*)-2.

Movie S4. The bond forming and breaking in the SZ616-catalyzed transformation of substrate (*R*)-1 to anti-Baldwin product (*R*)-3.

Overall, I commend the authors on a nice paper and look forward to seeing it in print.

Response: Thank you again!

Reviewer #4 (Remarks to the Author):

Overall, I believe these structures have not been modelled fully and would recommend the authors to revisit the structures and take more care to address the difference density that is present in the structures. I have made a list of the specific issues I have found within each structure.

Response: We thank the reviewer for the detailed comments. The following is our point-by-point response to his/her specific comments.

Of particular concern are the very high B-factors of the ligand molecule in 7XEE which much of the paper hinges on. I re refined the ligands against the mtz provided and remade the ligand and restraints and the B-factors were much lower. There is also a possibility of an alternate conformation of the ligand based on the presence of difference density when a single ligand conformation was modelled (see pictures in attachment).

Response: Thanks for your advices. In the revision work, we refined the CXI ligands with remade ligands and restraints in chains A and B. The resultant B-factors are *ca.* 31 Å² in chain A and *ca.* 25 Å² in chain B, which are much lower than the original ones (60~100 Å²). Meanwhile, the real-space correlation coefficient (RSCC) for the two ligands increased from 0.72 to 0.83 in chain A, and from 0.78 to 0.89 in chain B, respectively. While most regions of the density map in chain A can be fitted with CXI ligands, there are still unmodeled difference map peaks (**Figure R1a**). In contrast, CXI ligand in chain B is well fitted with the density map (**Figure R1b**), suggesting that ligand modelling based on the density map in chain B is reliable.

Figure R1 Modeling of each CXI ligand with one conformation, 2Fo-Fc map (blue), Fo-Fc map (green) and difference map peak (positive, magenta)

To handle the presence of difference density, we added alternative conformations of the ligands into the difference map in chain A (**Figure R2**). The occupancies of the two conformers are 0.55 and 0.45, respectively, and the corresponding B-factors are *ca.* 24 Å² and *ca.* 25 Å², respectively. The RSCC values for both conformers improved to 0.86 compared to the single conformer (0.83). Therefore, following the reviewer's suggestion, we refined the PDB structure with two conformers of CXI ligand in chain A, while maintaining one conformer in chain B.

Figure R2 Modelling of CXI ligand with two conformations, 2Fo-Fc map (blue), Fo-Fc map (green) and difference map peak (positive, magenta)

Structure 7VWD

Difference map peaks >5 sigma

- Number 2 and 6 look like a PED molecule not the waters that have been modelled

Response: Thanks for checking. After refinement, the difference map peaks for number 2 and 6 were eliminated. Please see the refined structure file.

- Number 4 should have a coordinating water added

Response: Thanks, the water molecule has been added.

- Number 10 should have a water modelled

Response: Thanks, it has been done.

Chain C has clear density for residues before residue number 5, up to residue -2 can easily be modelled.

Response: Following the suggestion, we have recruited the corresponding residues in chain C up to residue -2. Likewise, we also checked the other three chains, and modeled the missed residues in chain D up to residue -2 in terms of density. Please see the refined structure file.

Glu138 in chains A, C and D are poorly modelled to the density.

Response: Thanks, we have justified the Glu138 model based on the density map accordingly.

Leu147 backbone is out of the density

Response: Thank, it has been corrected.

The rotamer outliers for chain D are much higher compared to other chains, should be addressed.

Response: Compared the other chains, the higher rotamer outliers for chain D may be caused by the poor density of the particular region (residues 12 to 16). After removal of the modeling residues 12 to 16, rotamer outlier for chain D is reduced from 1% to 0 (see details in the validation report file).

Structure 7XEE

The B-factors for both the CXI ligands are very high, ~60 and ~100 in chains A and B, respectively. This is especially troubling as the B-factors of the surrounding residues are around 15. I would suggest the authors re refine their ligand with a lower occupancy to address this problem. The ligands also have a very high Z score for both bond lengths and angles.

Response: On the one hand, the problem of high B-factor values were successfully addressed by adding alternative conformations in chain A and restraints as described above (see Figures R1 and R2). On the other hand, the Z score of ligands was reduced to a reasonable level by optimizing the bond lengths and bond angles (see details in the validation report file)

Again, a number of difference peaks at 5 sigma or above should be addressed, some look like potential metal binding sites based on the coordination surrounding the peaks.

Response: Thanks, to erase the mentioned difference peaks, waters were recruited. In addition, some certain difference peaks that are not suitable for waters were considered to be metal ions. We have tried a series of ions, including Ca²⁺, Mg²⁺, K⁺, etc, and determined that Na⁺ would be optimal based on the metal-ligand distance, geometry, as well as occupancy.

Ser37 in chain B should have a second conformation modelled.

Response: Thanks, it has been added accordingly.

Ala12 is overlapping with a water molecule, this should be addressed.

Response: Thanks for the suggestion. We have removed the disturbed water molecules and re-modeled Ser12 in the corresponding density map.

Structure 7XEF

Incorrectly modelled side chains

- Arg9 – chain A
- Glu138 – chain C and D
- Leu147 – Chain C
- Ser111 – Chain D – Add alternative conf
- 121Glu – Chain D- Add alternate conf
- 78Met – Chain D – Add alternate conf
- 49Glu – Chain D – Add alternate conf

Response: Have been corrected accordingly. Moreover, we have checked the whole structures throughout, an alternative conf for Ser111 has also been added in chain A.

There is density in chain B for residues before number five up to -3 of the tag.

Response: Thanks, the mentioned residues have been added.

Structure 7VX2

There is density in chain C and D for residues before number five up to -3 of the tag. Currently waters are filling the density.

Response: Thanks for the suggestion. After removal of the disturbed waters, we noticed that the residue -3 in chain C, and residues -2 and residue -3 in chain D were not able to be modeled due to the poor density. Therefore, we modelled the corresponding residues from 5 to -2 of the tag in chain C, and residues from 5 to -1 of the tag in chain D.

There are a good number of difference density peaks around 5 sigma that should be addressed within the structure.

Response: Thanks, it has been addressed by recruiting waters.

Arg99 in chain C has an incorrect rotamer, with the NE atom facing away from Asp101

Response: The side-chain of Arg99 was refined accordingly, the NE atom now orientates to Asp101.

There are unmodelled blobs present in the active site of chain C that should be addressed.

Response: Thanks, the mentioned blobs are now modeled by adding one PEG molecule and one water molecule in the active site of chain C. After checking the other chains, this problem was also found in chain A, an EDO molecule was therefore added in the active site of chain A to occupy the blobs.

Arg99 in chain D is not interacting with Asp132 as I would expect.

Response: Thanks, the side-chain of Arg99 is now interacting with Asp132 by refinement.

Structure 7VMW

There are over 50 difference map peaks around 5 sigma which could and should be addressed.

Response: Thanks, we have done a full inspection of the structure, and found totally 56 difference map peaks. Among them, 34 water molecules and 3 ligands (one GOL and two EDO molecules) were employed to occupy the corresponding densities. And the rest of the difference map peaks were occupied by particular residues (e.g., Lys4, Glu45, Gln121 and Arg148) after refinement.

Gln121 is modelled incorrectly

Response: Thanks, an alternate conformation of Gln121 have been modeled in chain A and chain C.

PEG 3 in chain G should be remodeled into the difference density present and possibly has Asp101 modelled with an alternate conformation

Response: Following the suggestion, we have remodeled PEG 3(i.e., PEG 201 in chain C) into the

density. However, when modeling Asp101 with an alternate conformation, the occupancy of the alternate conformer is as low as 0.2, and the alternate conformer cannot be well fitted to the density (**Figure R3a**), suggesting that there may not exist additional conformation for Asp101. We therefore kept Asp101 in one conformation and instead modeled the water molecule into the density (**Figure R3b**) (including the density maps located between ligands and Asp101 in other chains), which is consistent with other reported LEH structures (PDB ID 4XBY; PDB ID 4XDW; PDB ID 4XDV; PDB ID 5CK6).

Figure R3 Modeling of an alternate conformation of Asp101 (a) or one water molecule (b)

Met78 on chain D should have an alternate conformation added.

Response: Thanks, it has been added accordingly. Likewise, an alternate conformation of Asp33 has also been modeled in chain C.

Density is present to model residue 4 in chain A.

Response: Residue 4 has been modeled in chain A now. Overall, we much thank the reviewer for his/her detailed suggestions and comments. We have improved all the five structures accordingly. The corresponding structural files, including atomic coordinates (.pdb format), reflection data (.mtz format) and the validation reports (.pdf format), are provided and described below.

SZ611-7VWD.pdb. The crystal structure of SZ611.

SZ611-7VWD.mtz. The reflection data of SZ611.

SZ611-7VWD-validation report.pdf. The validation report of SZ611.

SZ612-7VX2.pdb. The crystal structure of SZ612.

SZ612-7VX2.mtz. The reflection data of SZ612.

SZ612-7VX2-validation report.pdf. The validation report of SZ612.

SZ616-7VWM.pdb. The crystal structure of SZ616.

SZ616-7VWM.mtz. The reflection data of SZ616.

SZ616-7VWM-validation report.pdf. The validation report of SZ616.

SZ611-CXI-7XEE.pdb. The complex structure of SZ611-substrate 59.

SZ611-CXI-7XEE.mtz. The reflection data of SZ611-substrate 59.

SZ611-CXI-7XEE-validation report.pdf. The validation report of SZ611-substrate 59.

SZ611-D0I-7XEF.pdb. The complex structure of SZ611-product 46.

SZ611-D0I-7XEF.mtz. The reflection data of SZ611-product 46.

SZ611-D0I-7XEF-validation report.pdf. The validation report of SZ611-product 46.

REVIEWER COMMENTS

Reviewer #1 (Remarks to the Author):

The Authors have carefully addressed most of my comments. The manuscript has been changed accordingly or adequate replies to my questions have been provided.

However, concerning my last comment about the sentence at p. 17, lines 273-276 ("Two other essential mutations, Y53F and N55A, have well-defined electron density maps that together expand the hydrophobic local space, thereby excluding water molecules from the catalytic pocket, which would otherwise lead to 1,2-diol formation."), differently to what stated by the Authors in their replies, the text is unchanged. I would be happy with a rephrasing (or a suitable justification for not doing it is). Therefore, I suggest the Editor to accept this manuscript after minor revisions.

Reviewer #3 (Remarks to the Author):

The authors have satisfactorily revised the manuscript and it is now ready to be published in my opinion. Great job!

Reviewer #4 (Remarks to the Author):

Having look through the reworked structures I am now happy with the models accompanying this paper and believe they are ready for publication.

REVIEWERS' COMMENTS

Reviewer #1 (Remarks to the Author):

The Authors have carefully addressed most of my comments. The manuscript has been changed accordingly or adequate replies to my questions have been provided.

However, concerning my last comment about the sentence at p. 17, lines 273-276 ("Two other essential mutations, Y53F and N55A, have well-defined electron density maps that together expand the hydrophobic local space, thereby excluding water molecules from the catalytic pocket, which would otherwise lead to 1,2-diol formation."), differently to what stated by the Authors in their replies, the text is unchanged. I would be happy with a rephrasing (or a suitable justification for not doing it is).

Therefore, I suggest the Editor to accept this manuscript after minor revisions.

Response: Thank you very much! We have modified the mentioned sentence as "Two other essential mutations, Y53F and N55A, have well-defined electron density maps that together expand the hydrophobic local space, thereby leading to the preference of intramolecular cyclization reactions." It is also highlighted in the revised manuscript. Thank you again!

Reviewer #3 (Remarks to the Author):

The authors have satisfactorily revised the manuscript and it is now ready to be published in my opinion. Great job!

Response: Thank you very much!

Reviewer #4 (Remarks to the Author):

Having look through the reworked structures I am now happy with the models accompanying this paper and believe they are ready for publication.

Response: Thank you very much!